# Advancing Equity through Effective Youth Engagement in Public Health to Operationalize Racism as a Public Health Crisis: The Flint Public Health Youth Academy Model

Kent D. Key [1,2,*] , Kayla Shannon [2], Everett Graham [2], Cruz Duhart [2], Tomás Tello [2], Cole Mays [2], Christian Mays [2], Tyshae Brady [2], Jasmine Hall [2], Kahlil Calvin [2], Courtney Blanchard [1], Vanessa de Danzine [2] and Sarah Bailey [2]

1   Charles Stewart Mott Department of Public Health, College of Human Medicine, Michigan State University, Flint, MI 48502, USA
2   Flint Public Health Youth Academy, Flint, MI 48502, USA; cruz.malik.d@gmail.com (C.D.); tjoneslove453@gmail.com (T.B.); jasmhall2@gmail.com (J.H.); kcalvin@msm.edu (K.C.); secogeneral@yahoo.com (V.d.D.); baileysarah61@yahoo.com (S.B.)
*   Correspondence: keykent@msu.edu

**Abstract:** Background: The underrepresentation of BIPOC youth in the fields of public health, medicine, and research may be a factor contributing to the disproportionate rates of health disparities in BIPOC communities. In 2004, the Sullivan Commission on Diversity in the Healthcare Workforce, commissioned by the White House and led by the United States Health and Human Services, recommended efforts to increase the number of minority professionals in the aforementioned fields as necessary for addressing racial and ethnic health disparities. More recently, over 240 municipalities in the United States have declared "racism a public health crisis". This national declaration links racism directly to public health disparities, thus calling for a public health response. The Flint Public Health Youth Academy (FPHYA) provides an effective model of youth engagement steeped in Equity, Diversity, and Inclusion (EDI). FPHYA was created based on a dissertation study designed to explore the motivators for engagement of African American and other minority students into careers in public health and its six recommendations. Methods: The FPHYA Model described in this article uses a case study of the Flint Water Crisis to assess and explore effective youth engagement models for public health. This model is rooted in the Continuum of Community Engagement and Youth Empowerment Theory and explores FPHYA's contribution of youth voice in operationalizing racism as a public health crisis.

**Keywords:** public health; racism; youth engagement; youth empowerment; social justice





## 1. Introduction

The Case:

On 17 February 2017, the world witnessed the outcome of an investigation by the Michigan Civil Rights Commission focused on the city of Flint, Michigan. Three hearings later, and after reviewing a plethora of evidence and facts, the Commission concluded that the Flint Water Crisis (FWC) was an outcome of racism (systemic, environmental, and structural) [1]. The story of the Flint Water Crisis began after the Governor of the State of Michigan appointed an Emergency Manager (EM) to make all final decisions regarding the City of Flint, usurping all local elected officials' voices in decision making [2]. Such decisions, including the refusal to pay USD 150 per day for anticorrosion control of the water, changed the lives of Flint residents for generations to come.

The decisions made by the EM impacted all aspects of Flint residents' life, even industry. Months after the initial water source switch, General Motors, one of the leading automobile industry leaders in the United States, observed the impact the water switch was having on production. The water was damaging the materials in their Flint plants,

causing harm to metals, chemicals, and other needed parts. The State of Michigan allowed General Motors to disconnect from Flint Water and return to the Detroit Water Authority water. However, Flint residents were not given the same consideration and were forced to drink, bathe, and cook with water that damaged automotive materials [3]. This majority-African-American city experienced a threefold crisis relating to (1) democracy, (2) water, and (3) public health, which was due to a failure of government on all levels rooted in racism.

## 2. Background

One of the impacts of the Flint Water Crisis was its negative impact on youth. Studies suggest that exposure to lead impacts the cognitive and behavioral domains of children and youth [4]. Media outlets (local, national, and international), for over a year, broadcasted stories of the impact of lead exposure on Flint youth. Over decades, a plethora of scientific articles across disciplines shared the irreversible impact that lead has on developing youth [5–9]. Flint youth were now branded as damaged on both a national and global stage.

In 2014, Dr. Kent Key, Flint resident, completed his dissertation study examining the motivators for engaging African American students into careers in public health. The impetus for this study was the glaring disparities across health indicators (cancer, stroke, diabetes, maternal mortality, and infant mortality) where African Americans were disproportionately impacted. This was coupled with the gross underrepresentation of African Americans in the field of public health. The result of this qualitative study yielded six recommendations [10]. Those six recommendations were used as the foundation to create the Flint Public Health Youth Academy (FPHYA). Dr. Key worked for two years building a curriculum for FPHYA. During this time, the Flint Water Crisis was in the national news, and the lead's impacts on Flint youth were front and center. As a result of lead exposure, media outlets reported that Flint youth would not be able to excel academically and would have behavioral issues. Dr. Key utilized strategic initiative funding from his Robert Wood Johnson Foundation Culture of Health Leaders Fellowship and launched the Flint Public Health Youth Academy. FPHYA is comprised of three programming domains: (1) the Community Assessment Domain (Photovoice, community dialogues); (2) the Learning Academy Domain (FPHYA curriculum, summer camps, youth summits); and (3) the Advocacy/Policy Domain (public comment, government engagement, policy development) (Youth Academy—Community Engagement Studio of Flint (cestudioflint.org). In addition, FPHYA has created effective virtual programming via the "Youth Perspectives" talk show on the FPHYA Facebook and YouTube pages. This talk show has view counts as high as 8K.

In 2020, a national move across the United States began to occur, declaring racism a public health crisis. In June 2020, Genesee County, MI (the home of Flint), made a county-wide declaration. Dr. Kent Key, local public health scientist and Flint resident, authored that resolution with local activist Nayyirah Shariff [11]. Over 240 municipalities declared racism a public health crisis across the country [12]. In January 2021, *The Nation's Health*, the national newspaper of the American Public Health Association (APHA), identified Dr. Key and the work in Genesee County as one of three municipalities leading this work [13]. This resolution lifted racism as a negative contributor to public health, suggesting the need for a public health response. In 2021, Dr. Key and his FPHYA team recognized the need for youth voice and perspective in operationalizing racism as a public health crisis and held seven youth dialogues, engaging eighty-one youth participants across six zip codes to gain youth perspectives and thoughts concerning racism and how to dismantle it, yielding a report published in January 2022 (FPHYA Dialogue Session Report (beyondrhetoricmovement.com, accessed on 1 March 2024).

The intersection of the Flint Water Crisis and declarations of racism as a public health crisis has yielded the Flint Public Health Youth Academy (FPHYA), a remarkable youth program in Flint, MI. **The goal of this article is threefold:** (1) to respond to the six recommendations from the Key (2020) article [10]; (2) to propose a new youth engagement model, the FPHYA Model, to evaluate the aforementioned six recommendations; and (3) to present this model as an Equity, Diversity, and Inclusion (EDI) approach for engaging youth

voices in operationalizing racism as a public health crisis by highlighting the process and outcomes of the FPHYA youth dialogue on racism.

While we understand that there are many approaches to youth engagement, and no one model or framework is absolute, we posit that this model can serve as one of the many tools for engaging youth into the public health profession and the effort to operationalize racism as a public health crisis from a youth perspective. We are not reporting research; we are assessing a conceptual model/framework based on science and real-life application for youth programming focused on a public health/social justice issue (racism as a public health crisis).

## 3. Methods

In the development of our model, we examined 45 articles from the literature on youth engagement including research studies, reviews, theories, models, and frameworks. We identified 15 articles that focused specifically on at least one or more of the following criteria: (1) BIPOC youth mentorship, (2) pipeline/pathway to careers programs and (3) public health workforce, (4) youth empowerment. This review helped us to identify the domains needed in a comprehensive model to effectively engage youth into the field of public health. Furthermore, we wanted a model to address racism and the inequities and negative disparities it produces in the lives of BIPOC communities nationally and globally. Finally, we centered our efforts on Youth Empowerment Theory and the Continuum of Community Engagement.

## 4. Review of the Literature

There remains a crucial need for a diversified public health workforce. African Americans and other people of color are disproportionately underrepresented in the fields of public health, medicine, and research, as shared in a national report by the Sullivan Commission [14,15]. The presence of African Americans and other people of color as medical and public health professionals may positively impact trust in healthcare and improve the level of culturally competent care for minority communities [10,16]. This is supported by data that show that minority health professionals are more likely to serve populations of their own ethnicity [14]. Creating a diverse public health and medical workforce is part of a national strategy to address racial and ethnic health disparities [17].

Pipeline and mentor programs are mechanisms that can be utilized to develop future public health and medical professionals, especially those from BIPOC communities. Over the last few decades, mentoring programs have been a strategy for the engagement of and intervention for "at-risk" youth [18]. A systematic review of youth mentoring programs from 1975 to 2017, including 70 mentoring studies with a total sample size of 25,286 youth, concluded that mentoring significantly impacts the trajectories of youth [19]. Reyes conducted a systematic review of 24 studies and concluded, in a study of 40,000 plus youth, that 76% of the mentees receiving services were youth of color. This review concluded that mentoring positively impacts a range of needs, including psychological, social, and behavioral needs, while serving as a low-cost intervention [19], and promotes positive identification and cognitive and social development [20].

Mentoring is an effective tool for youth development and career exploration. Efforts to promote science, technology, engineering, and mathematics (STEM) have gained popularity in the last few decades. Coupled with mentoring, STEM is an effective way to expose youth to career fields while providing didactic and experiential learning [21]. Lyons further shares that STEM youth mentoring programs utilize the Youth Science Pathway, which includes a three-tier model: (1) discover; (2) explore; and (3) pursue. This model provides a method of engagement for youth and an approach for youth programmers. An early introduction to careers in public health is critical for career trajectory and career attainment [10].

Youth mentoring programs have been successful at applying a social justice lens to programming. In 2017, Albright et al. [22] conducted a literature review of youth mentoring programs that focused on social justice. They concluded that there was a need for more

attention to social justice principles to be embedded in the design and implementation of youth programming. Anchoring youth engagement in principles of social justice is paramount for producing a generation of transformative thinkers that will contribute to the solutions to inequities and disparities that disproportionately impact minority and low-income communities. Thus, social justice engagement and critical consciousness education (a fundamental understanding of oppressive social elements and hierarchal structures) are key elements [22]. Applying a social justice framework may be successful at reaching marginalized at-risk youth who are most disproportionately impacted by structural oppression and systems of hierarchy.

Efforts to operationalize the declaration of racism as a public health crisis should include an Equity, Diversity, and Inclusion approach (the intentional inclusion of youth voices and perspectives) with emphasis on youth from BIPOC and other marginalized communities. Racial justice, anti-racism, and equity work are synonymous with mobilizing communities to address the historical and generational impact of racism. Efforts to operationalize over 240 policy declarations of racism as a public health crisis must model a true EDI approach. As part of that approach, many diverse groups of individuals must be included in the development of solutions. Racism is an interdisciplinary problem that contributes to racial and ethnic inequities and disparities, impacting BIPOC and low socio-economic communities disproportionately [23]. Just like the public health workforce, diversity is critical if we are to collectively create a public health response to racism. Thus, in the spirit of EDI, youth must have an active role, a respected voice, and a physical presence in this process.

## 5. FPHYA Model

We present the FPHYA Model as a framework for providing mentoring, social justice, and career exploration programming to young people while priming them to be the next generation of public health professionals. The *Journal of African American Studies* released a publication in 2020 sharing six recommendations for building a youth empowerment pathway to careers model to engage African American students into careers in public health [10]. This study was guided by the Feedforward Model. This Feedforward Model is a decision-making framework that explores three components: (1) forethought, (2) planning, and (3) the state of being proactive [24]. This model was used to assess African American public health professionals' academic and career trajectories and the factors contributing to that decision-making process. This study yielded the follow six recommendations for effective youth engagement into public health: (1) visibility/branding; (2) caped warrior/celebrity model; (3) framing public health as social justice; (4) role models/mentorship; (5) packaging public health with behaviors; and (6) public health summer employment opportunities [10]. Those six recommendations were used to create the FPHYA Model (see Figure 1).

The FPHYA Model is supported by **Youth Empowerment Theory (YET)** (see Figure 2) and guided by the **Continuum of Community-Engaged Research Model (CCEnRM)** (see Figure 3). Youth empowerment programs anchored in Youth Empowerment Theory aim to incorporate highly participatory youth-driven processes [25]. YET underscores the equitable inclusion of youth not as program participants only, but also in the design and implementation of youth programming. Furthermore, YET fosters a psychological empowerment process comprised of three components/domains: (1) the Intrapersonal Component (self-esteem, leadership efficacy, and civic efficacy); (2) the Behavioral Component (leadership behavior, community engagement, school engagement); and (3) the Interactional Component (adult mentors, adult resources, and resource mobilization) [26]. The CCEnRM takes its roots from the ladder of civic engagement [27]. This model provides three main domains: (1) the various destinations along the continuum of engagement, (2) contextual factors, and (3) equity indicators. This continuum underscores the importance of community engagement, but equally the importance of factors such as mutual benefit, decision making, power and control, history, trust, and transparency and how they contribute to how and what that engagement looks like and the impact of said engagement [27].

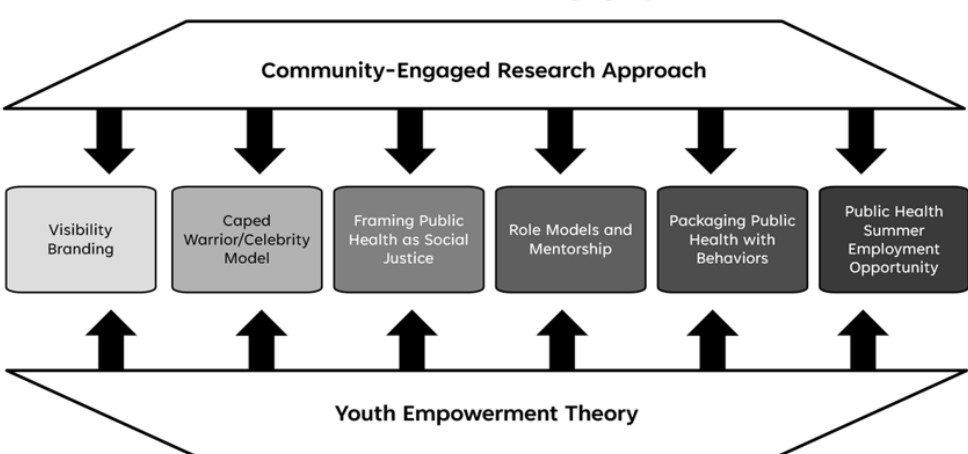

**Figure 1.** FPHYA Model.

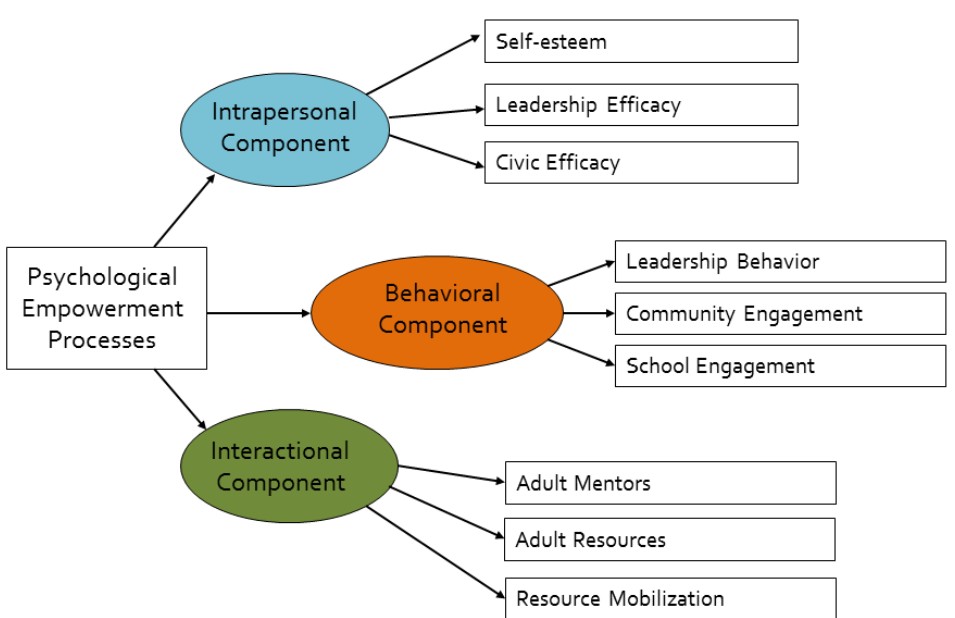

**Figure 2.** Youth Empowerment Theory (YET) [26].

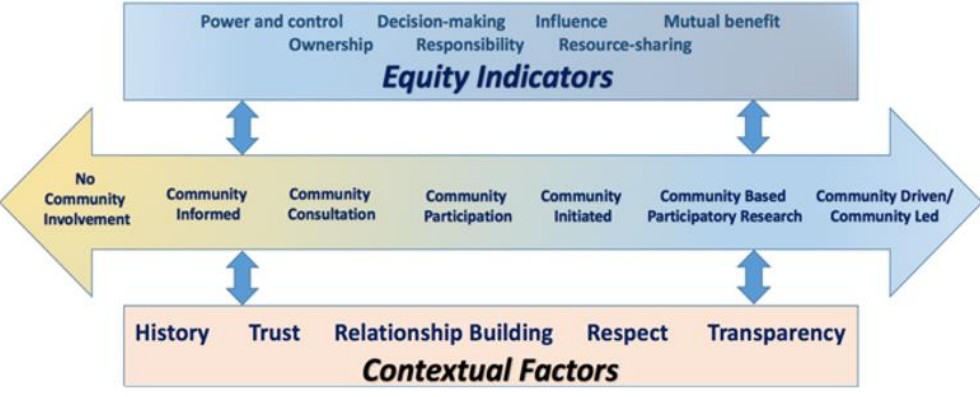

**Figure 3.** Continuum of Community-Engaged Research Model (CCERM) [27].

In efforts to operationalize racism as a public health crisis, the FPHYA Model comprising of YET and the CCEnRM was used to conduct youth dialogues with youth in Flint/Genesee County to explore racism as a public health crisis. The FPHYA Model incorporates the six recommendations from the 2020 Key article [10]: (1) visibility/branding; (2) caped warrior/celebrity model; (3) framing public health as social justice; (4) role models/mentorship; (5) packaging public health with behaviors; and (6) public health summer employment opportunities. We will access efforts made in these six domains to advance racism as a public health crisis in Flint/Genesee County.

**Visibility and branding.** Since 2018, the FPHYA has created visibility and branding that is distinct from that of other youth programs in the Flint/Genesee County community. FPHYA has partnered with over 50 local youth programs to provide distinct and unique competencies specific to public health. FPHYA has worked with local community school districts, church youth groups, fraternity/sorority youth mentor groups, and other non-profit entities to provide programming to over 3000 youth. Furthermore, FPHYA youth have presented their programs and findings at national scientific meetings such as the American Public Health Association and the Community-Based Public Health Caucus. This visibility has garnered on-going interest from academic institutions, philanthropic institutions, and state and local agencies wanting to partner with FPHYA interns as consultants for a host of projects including topics such as (1) COVID-19 messaging, (2) environmental health, and (3) mental health, to name a few. The successful visibility and branding of FPHYA provide community support for FPHYA to create a safe space for youth in Flint/Genesee County, Michigan, to have conversations about racism and to begin discussing solutions to address it.

**Caped warrior/celebrity model.** The FPHYA has engaged with various types of celebrities, whether pop-culture celebrities or celebrities in their respective fields. Jenifer Lewis, movie star and, most recently, TV star in the hit series *Blackish*, came to the city of Flint shortly after the onset of the Flint Water Crisis and met with the FPHYA and other youth groups in the city. Over time, a relationship was created, and Ms. Lewis, who is a national public spokeswoman advocating for mental health, specifically bipolar disorder, has made appearances on the FPHYA's "Youth Perspectives" talk show, focusing on mental health. Ms. Lewis is vocal on her social media platforms, which reach millions of viewers, as she continues to highlight racism and its impact on the Flint Water Crisis. FPHYA continues to partner with various celebrities to bring awareness regarding public health and social justice.

**Framing public health as social justice.** The FPHYA, since its inception, has linked current and relevant social justice issues to public health. The Social Determinants of Health are a staple in the FPHYA curriculum, and lessons on the linkage between the SDOH and health inequities and health disparities connect public health with many of the social injustices that youth see nationally. Racism and discrimination are SDOH. The FPHYA Model provides guidance to empower youth to address those issues. The declaration of racism as a public health crisis provided another opportunity for FPHYA to address and link a social justice issue (racism) to public health frameworks, approaches, and models. Furthermore, it provided an opportunity to use effective community engagement strategies like community dialogues to capture themes, ideas, and recommendations from local youth regarding racism through a public health lens.

**Role models and mentorship.** Embedded in the FPHYA Model and the YET are mentorship and role models. FPHYA incubates in-program participants with the opportunity for mentorship. In the FPHYA Flow of Engagement (mentorship is foundational), FPHYA youth interns are supported by FPHYA mentors in generating educational modules, youth engagement opportunities, writing grants, making presentations, and conducting policy and advocacy activities. FPHYA mentors consist of public health professionals, medical professionals, law enforcement professionals, mental health clinicians, community organizers, activists, and policy makers to name a few. More importantly, FPHYA mentors live in or work in the communities where FPHYA youth live. That visibility is vitally important to

mentorship. FPHYA interns who progress through the program return as FPHYA mentors supporting the next wave of youth interns (see Figure 4).

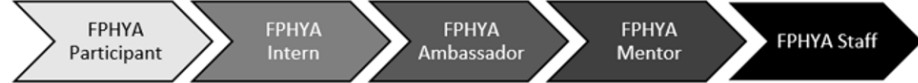

**Figure 4.** FPHYA Flow of Engagement.

**Packaging public health with health behaviors.** Health behaviors and self-accountability are key features in the FPHYA program. Public health issues like smoking, obesity, diabetes, and community violence often have health behavioral factors associated with them. FPHYA has been actively engaging youth in conversations related to these topics, and, more importantly, providing a platform for youth to share their perspective on these issues. During annual summer camp programming, FPHYA youth have launched several Photovoice projects highlighting these health behavior/health outcome concerns and have recommended policy solutions to support behavior change. More recently, FPHYA has launched a youth perspective health campaign on gun violence, examining behaviors, assessing laws and gun regulations, and looking at the role of self-accountability for both parents and youth.

**Public health summer employment opportunities.** Summer employment initiatives have been vital for engaging youth in public health and providing academic and professional experience for that trajectory. Initially, FPHYA programming provided summer employment during our summer camps. After the onset of COVID-19, FPHYA youth internships became year-round and not just for the summer. Youth interns supported by FPHYA staff pivoted from in-person programming to virtual programming, which resulted in the launch of the "Youth Perspectives" talk show. To date, we continue to provide monthly stipends to FPHYA youth for developing, facilitating, and evaluating programming initiatives/projects. Furthermore, youth interns from FPHYA are connected to a mentorship team who leverages their networks to provide additional opportunities for employment, internships, and scholarships for FPHYA youth.

## 6. Discussion

In our case study, we highlighted racism and its negative impact on an entire city caused by a political decision that created the worst modern-day public health crisis of our time. This case is a classic example of the impact racism has on the public health of BIPOC communities and the various and varied health inequities and disparities experienced by those populations. The message, declared all over the country, is clear: racism is a public health crisis. This crisis, framed in the context of public health, calls for a public health response to racism. The need for racial justice in health is evident. In addition, the need to diversify the public health workforce is on-going and important to national efforts to reduce racial and ethnic health disparities. Effective youth mentoring programs can play effective roles in early career exploration and attainment, centering on social justice (as suggested in the FPHYA Model). Combining mentorship with career exploration programs may be essential to BIPOC youth and their communities. These types of experiential learning opportunities can assist youth in creating networks to become effective contributors to public health and social justice issues relevant to our national and global communities. Furthermore, it introduces youth to career pathways to public health and other adjacent careers such as those in medicine, social work, policy, and other disciplines needed to address the social injustices and health disparities that result from inequities caused by systemic and structural racism. The FPHYA Model has worked for Flint and could possibly be a template to be tailored and scaled out nationally.

## 7. Future Direction and Implications

The FPHYA Model helps address the need for effective youth programming within a social justice context that can contribute to strategies to address racism. In addition, this model can be used to address other social justice issues rooted in health inequities and disparities. Finally, this model was created by academics, community partners, and youth and was revised with input from the broader community. More importantly, this model has been implemented by the FPHYA and has yielded effective engagement and programming for youth in Flint/Genesee County, Michigan.

Linking public health, disparities, and inequities to social justice serves as an effective catalyst for youth engagement. Historically, youth have been involved in most major movements that have made a critical impact in the United States, such as the civil rights and the women's liberation movements. Youth have significant roles in advancing causes, demonstrating, and mobilizing marches and other protests. Linking this type of youth synergy for the cause of social justice to critical public health inequities is an ideal method and approach for (1) introducing youth to careers in public health; (2) engaging youth in the civic process; (3) creating a platform for youth to form their own agendas; and (4) creating a template from which adults can learn and understand effective intergenerational engagement with youth, especially BIPOC youth, who often feel unseen and unheard.

Public Health:

The FPHYA Model has implications for engaging youth perspective to operationalize racism as a public health crisis. There is no one cookie-cutter template for addressing racism. It is a systemic and complex issue which will call for a systemic and complex response. With more than 240 declarations across the United States, this model can contribute to toolkits, strategies, and approaches that utilize public health approaches to address community and social issues such as racism. More importantly, the FPHYA Model is a public health model designed specifically for youth grounded in science and the theory of youth empowerment and guided by the Continuum of Community Engagement. These two foundational elements have been proven to be effective in the field of public health and beyond. Public health planners, programmers, and recruiters may learn effective youth recruitment and engagement strategies from the FPHYA Model. Given the need for more public health professionals in the workforce, and, more importantly, the need for BIPOC public health professionals, the FPHYA Model can contribute to a national strategy to advance public health by recruiting youth from the communities most impacted by health disparities. This early introduction to the field could be the catalyst for a youth career pathway to public health.

Policy:

The FPHYA Model is an effective strategy for engaging youth in a solution-focused environment. Engaging youth through a public health lens and utilizing community assessments/community dialogues and Photovoice methodologies creates a platform for youth to share their perspectives with local leaders, policy makers, and other relevant stakeholders. The FPHYA Model also underscores the importance of advocacy and community mobilization, which contribute to good civic engagement. Thus, it is critical that policy makers (1) engage consistently with youth; (2) create clear lines of bi-directional communication; and (3) ensure youth-friendly spaces and platforms. If we are to ensure that youth understand and participate in the civic process, which is central to how the country is governed, elected officials and other stakeholders should support youth programming that introduces youth to the policy-making process at all levels of government (city, county, state, and federal).

Equity, Diversity, and Inclusion (EDI):

As the United States continues to diversify and grow, attracting ethnicities from across the globe, the need for representation becomes critical. Advancing EDI should be a pathway to advance youth voice and perspective. The FPHYA Model ensures the EDI of youth in public health concerns, community issues, and social justice efforts. Youth who are often overlooked and usually not engaged have a lot to contribute. They see things differently,

they experience things differently, and they hear things differently. As leaders of the present and future, their input, suggestions, feedback, and recommendations are central to advancing our local, regional, national, and global communities. We cannot advocate for EDI with only adults in the room. When we speak of EDI, we must also have our youth in mind, and, more importantly, include them in the processes, strategies, and decisions we make.

## 8. Conclusions

It is our hope that this model will serve as a guide for youth-empowering programs, especially those serving BIPOC youth. We further hope that the value of mentorship is underscored as a critical component for effective youth engagement. In addition, this model may serve as a blueprint for recruiting and diversifying the public health work force with people representing BIPOC communities. This model can serve as a guide for researchers, public health professionals, youth programmers, civic leaders, and policy makers who desire to include a youth voice and youth perspective.

To the extent possible, this information can be shared with youth programmers and the public, policy makers, school officials, and leaders of various health initiatives to enhance their understanding of effective youth engagement. We also anticipate that the FPHYA Model will be adopted by universities, institutions, organizations, and agencies to effectively provide guidance and direction centering on youth to create transformative change communities across this country.

**Author Contributions:** K.D.K. and S.B. drafted and provided considerable revisions to this manuscript. K.D.K. and C.B. contributed to the conceptualization of the FPHYA Model. K.S., E.G., C.D., T.T., C.M. (Cole Mays), C.M. (Christian Mays), T.B., J.H., K.C., C.B. and V.d.D. contributed to writing and edits. All authors have read and agreed to the published version of the manuscript.

**Funding:** This submission was supported by National Institute of Health: K01MD015079; Community Foundation of Greater Flint: 208887.

**Institutional Review Board Statement:** Not applicable.

**Informed Consent Statement:** Not applicable.

**Data Availability Statement:** No data was created. Information about FPHYA can be obtained at www.cestudioflint.org (accessed on 1 March 2024).

**Acknowledgments:** The authors would like to thank representatives of the Flint community (especially Flint youth) for reviewing and providing feedback regarding the development of the FPHYA Model.

**Conflicts of Interest:** The authors declare no conflict of interest.

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
