# Peer review of "Advancing Equity through Effective Youth Engagement in Public Health to Operationalize Racism as a Public Health Crisis: The Flint Public Health Youth Academy Model"

_2673-995X, doi:10.3390/youth4010028_

Round 1
Reviewer 1 Report
Comments and Suggestions for Authors
Please find edits and suggestion within the attached PDF. Anchor the article in two of the frameworks included. There are too many included reflecting a lack of focus on the purpose of the article which should be used to make your selections

Comments on the Quality of English LanguageOnce major revisions are made, a thorough editorial review is in order
Author Response
Reviewer 1:
Thank you for your suggested edits. I have made the following changes to the manuscript:
- The word “was” was deleted right before the word “commissioned on line 9
- The word “addressing” was changed to “address” on line 11
- The words Equity, Diversity and Inclusion is no longer capitalized on lines 14-15
- MI is now spelled out to say Michigan on line 27
- Lines 62-64 were reworded for clarity
- The word “Domain” was added after “Community Assessment” on line 67
- Lines 104-108 were added to clarify the literature review

Reviewer 2 Report
Comments and Suggestions for Authors
Connecting the history of the FPHYA with the FWC makes sense, especially including that GM was allowed to switch water due to damage to materials. The article details how the FPHYA responded to recommendations in Dr. Key's 2020 article and how those recommendations were evaluated by the FPHYA model as well as how youth were empowered to dialogue on racism. As stated in the article, the assessment of the framework/model seems thorough. The studies cited appear relevant and supportive of the assessment. The FPHYA model is thoroughly detailed and evidence is provided to support the model via other processes.
Author Response
Thank you reviewer two for understanding that this was not an research study, so the usual research markers in a research article were not applicable. Yet, this was a conceptual framework/model submission. Thank you for your positive feedback. I have edited a few lines in the manuscript per other reviewers requests highlighted in yellow.

Reviewer 3 Report
Comments and Suggestions for Authors
Suggestions to the article.
This work is a descriptive set of models, which is carried out in a haphazard manner and without a logical structure from a scientific perspective.
The research problem is clearly defined, but it does not correspond to the subsequent assessments that make up the body of the work. The alleged valuation does not specify the indicators on which it will be carried out, nor does it present scientifically verified results.
The variables or dimensions of the study are missing. What is valued? About what? And what is it compared to?
The source review is related to the main topic of the work, however, since the evaluative approach provided for in the problem is missing, it is not relevant without the interpretation of why each source is chosen.
There is a lack of a methodological design, with a detailed description of the methodology, data collection, sample, procedure, material and recording system, data analysis, etc. An analysis of the data on the work is not specified.
Conclusions are not justified should be supported by evidence from the findings.
There is no adequate discussion of the results of the study, relating them to those obtained in previous works by other authors.
In short, the structure of the document does not correspond to that of a scientific article. It presents extensive unorganized material in which the reader is lost in incomprehension of the text.
Author Response
Thank you for your thorough review. This manuscript article submission is not a research study. Thus it does not meet the standards of traditional research. It is a conceptual framework model/theory submission. The idea is to show how the conceptual framework that guides the Flint Public Health Youth Academy was developed. It (the model) is supported by previous work such as Youth Empowerment Theory and the Continuum of Community Engagement. In addition is supported by the recommendations from a dissertation study that was published as original research. That article met the criteria of research such as research question, research design and research results.
The following edits were made to the manuscript to add clarity.
- The word “was” was deleted right before the word “commissioned on line 9
- The word “addressing” was changed to “address” on line 11
- The words Equity, Diversity and Inclusion is no longer capitalized on lines 14-15
- MI is now spelled out to say Michigan on line 27
- Lines 62-64 were reworded for clarity
- The word “Domain” was added after “Community Assessment” on line 67
- Lines 104-108 were added to clarify the literature review

Round 2
Reviewer 1 Report
Comments and Suggestions for Authors
All major revisions have been addressed
Comments on the Quality of English LanguageQuality of English has greatly improved
Reviewer 3 Report
Comments and Suggestions for Authors
Taking into account that it is a theoretical work, characteristics such as: logical structure and coherence that give reason to the arguments are appreciated, presenting conclusions that support the information for future decisions in the area of work.